# Mothers at risk of postpartum depression in Sri Lanka: A population-based study using a validated screening tool

Therese Røysted-Solås[1]*, Sven Gudmund Hinderaker[1], Lasantha Ubesekara[2], Vijitha De Silva[3]

1 Department of Global Public Health and Primary Care, University of Bergen, Bergen, Norway, 2 Ministry of Health, Southern Province, Galle, Sri Lanka, 3 Department of Community Medicine, University of Ruhuna, Galle, Sri Lanka

* therese.solas@student.uib.no

**Data Availability Statement:** All relevant data are within the paper and its Supporting Information files.

## Abstract

### Background

Postpartum depression is an important public health concern. The prevalence of postpartum depression is estimated to be 18% worldwide. The purpose of this study was to estimate the prevalence of mothers at risk of postpartum depression in Sri Lanka and to investigate its associated risk factors.

### Methods

This was a cross-sectional study conducted among 975 mothers in Galle district, Sri Lanka. The prevalence of mothers at risk of postpartum depression was assessed using the Edinburgh Postpartum Depression scale (EPDS) which has been validated for screening for mothers at risk of postpartum depression in Sri Lanka with a cut-off score 9 or more. Prevalence was estimated using a cut-off 9 or more, 10 or more, 11 or more and 12 or more to assess the difference in prevalence using unvalidated cut-offs for screening. Data from routine records on pregnancy, delivery and postnatal care was collected to investigate possible predictors of EPDS score 9 or more (risk of postpartum depression). Univariate and multivariable logistic regressions were performed to identify risk factors for EPDS score 9 or more (risk of postpartum depression).

### Results

The prevalence of mothers with EPDS score 9 or more was found to be 9.4% (95%CI: 7.8–11.4); EPDS score 10 or more was 5.6% (95%CI: 4.4–7.3). EPDS score 9 or more (risk of postpartum depression) was associated with the following risk factors: Former history of mental illness (aOR 32.9, 95%CI: 7.9–136.2), high maternal age 30–39 (aOR 2.2, 95%CI: 1.3–3.8), BMI 25.0–29.9 (aOR 2.6, 95%CI: 1.5–4.5), hypertension (aOR 3.6, 95%CI: 1.2–10.9) and newborn death (aOR 28.9, 95%CI: 4.5–185.1). One in five women reported thoughts of self-harm.

**Funding:** The authors received no specific funding for this work.

**Competing interests:** The authors have declared that no competing interests exist.

## Conclusion

Around one in ten mothers in Sri Lanka experience symptoms of postpartum depression, highest risk among mothers who reported former history of mental illness and newborn death. The prevalence estimates were lower with a higher cut-off for screening and this highlights the importance of using the validated cut-off for screening in future studies on postpartum depression in Sri Lanka. Mothers at increased risk should be identified in antenatal care and are important targets of referral.

## Introduction

Depression after giving birth is considered a serious public health problem worldwide, and the global prevalence is estimated to be 18% [1]. Studies suggest that postpartum depression (PPD) not only affects the wellbeing of postpartum women, but also the mother-infant interaction [2]. There is a compelling body of evidence that PPD may influence the social, cognitive and emotional development of the child and infant growth [2].

PPD and its associated risk factors are well studied in high-income countries but data from low- and middle-income countries is scarce [3]. However, research suggests there is a higher prevalence of PPD in low- and middle-income countries compared to in high-income countries [3], which aligns well with low socioeconomic status as a known risk factor [2]. It is widely recognized that PPD is not attributable to one single cause, but rather influenced by complex interactions between environmental, psychosocial and biological risk factors [2]. Several risk factors associated with PPD have been investigated in the past; mothers with a former history of mental illness seem to be at higher risk of PPD [4], while maternal age as a risk factor have shown conflicting results [5].

Screening for and managing PPD is included in *The Global Strategy for women's, children's and adolescent's health (2016–2030)* by the World Health Organization [6]. Several tools to screen for PPD have been used, and this is reflected in the diversity in reported prevalence in different countries [5]. The Edinburgh Postnatal Depression Scale (EPDS) is a 10-item questionnaire and the most widely used tool globally to screen for mothers at risk of PPD [2]. Each answer is scored 0–3 and a higher sum of scores indicates more symptoms of depression.

Originally, the EPDS was developed for screening for PPD in English-speaking women and a cut-off 10 or more for "possible depression" and a cut-off 13 or more for "probable depression" was suggested [7]. Since then, the EPDS has been translated to and validated in many languages using different cut-offs for screening in different countries. It is important to emphasize that most studies on PPD do not report on the real prevalence of PPD, but rather the prevalence of mothers at risk of PPD screened with the EPDS. In 2006, Matthey et al. addressed the increasing use of unvalidated EPDS cut-off scores in studies on PPD [8]. Furthermore, he stated that a difference of just one point in cut-off score can have a substantial impact on estimates of mothers at risk of PPD [8]. Optimal psychometric properties are important for assessing prevalence rates and the overall burden of PPD in countries [8].

A potentially alarming symptom of PPD are thoughts of self-harm, which can be identified by the EPDS item 10 "the thought of harming myself has occurred to me". Thoughts of self-harm in pregnancy is not uncommon [9]; studies suggest that attention should be given to mothers with thoughts of self-harm as it can indicate more severe depressive symptoms and therefore increased risk of negative outcomes for mother and child [9]. Furthermore,

postpartum mothers who reported thoughts of self-harm had increased risk of somatic and psychiatric disease seven years after delivery, which could not be fully attributed to depression [10].

The reported prevalence of PPD in South-Asia has ranged from 5–49% [1]. Sri Lanka is a lower middle-income country and has been considered a role model for maternal health promotion and for reducing maternal deaths. In Sri Lanka the Sinhala version of the EPDS was introduced and included as a part of postnatal care in 2012 [11] and it has been validated for screening for mothers at risk of PPD with a cut-off 9 or more [12]. In 2011, a cross-sectional study representing eight provinces in Sri Lanka estimated that 27.1% of the mothers had an EPDS score 10 or more [13]. A recent study from 2017 in Galle, Sri Lanka, found that 7.8% of the mothers had an EPDS score 10 or more [14]. To this date there are no published studies available on the prevalence of mothers at risk of PPD in Sri Lanka using the validated cut-off for screening EPDS score 9 or more. Using a higher cut-off may lead to an underestimation of mothers at risk of PPD in Sri Lanka. Therefore, in our study we aimed to: 1) estimate the prevalence of mothers at risk of PPD at postnatal clinics in Galle using an EPDS cut-off 9 or more 2) evaluate the difference on prevalence estimates of mothers at risk of PPD using several cut-offs 3) investigate risk factors associated with EPDS score 9 or more (risk of PPD) among mothers in Sri Lanka.

## Methods

### Study design and study setting

A cross-sectional study was conducted among mothers who had delivered from 1st January 2019 to 31st December 2019 in Bope-Poddala, Medical Offices of Health (MOH) Division, Galle district, Sri Lanka. With a population of over 71 000, Bope-Poddala is a semi urban MOH division in Galle district. All three main population groups of Sri Lanka live there: Sinhalese, Tamils and Muslims. Here preventive care services are provided through 20 divisions of the MOH. Antenatal and postnatal care is provided to the mothers through five maternal and child health field clinics. In Sri Lanka the antenatal care attendance is very high, and nearly 100% of deliveries occur in health facilities. Maternal mental health services are provided during regular visits in the clinics by a medical officer of mental health.

### Participants

In Bope-Poddala, around one thousand births take place each calendar year. The sample size for the postpartum mothers to be included in the study was estimated using OpenEpi.com calculator [15]. Assuming that the prevalence of mothers at risk of PPD in Galle would be around 18% [1] and aiming at estimating the proportion with a precision of +/- 2.5%, the required sample size for our study was estimated to be n = 907. Therefore, all the postpartum mothers who delivered from 1st January 2019 to 31st December 2019 were included in the study. Mothers with incomplete EPDS records were excluded from the study.

### Study tool

In Sri Lanka, all mothers are routinely screened once at the clinics for depressive symptoms four to six weeks following delivery using the Sinhala or Tamil version of Edinburgh Postpartum Depression Scale (EPDS). An EPDS score 9 or more has been validated for screening for mothers at risk of PPD with the Sinhala version of the EPDS, with a sensitivity and specificity of 89.9% and 78.9% respectively [12]. The Tamil version of the EPDS is used for screening in

the Tamil population and has been validated for screening in Tamil Nadu, India using the same value, EPDS score 9 or more, with a sensitivity of 94.1% and a specificity of 90.2% [16].

In current practice, the EPDS is filled on paper forms by the mothers at their postnatal visit at the clinic and is then collected and kept by the midwives. If the mother cannot read Tamil or Sinhala the EPDS is completed by the midwife through an interview with the mother. Routinely, mothers with an EPDS score 9 or more are referred to a medical officer of health at the clinics who then refer the mother to their designated medical officer of mental health (MOMH). MOMHs manage some of the cases, and if necessary, refer the mother to psychiatrist care at the nearest hospital. In severe cases the medical officer of health refers directly to a consultant psychiatrist. MOMHs and psychiatrists diagnose mothers with PPD using the ICD-10 classification for diagnosis, usually through clinical interview with the mother.

## Data collection

All postpartum mothers have EPDS records and routine records on pregnancy, delivery and postnatal care in the B part of the pregnancy record which are kept in the clinics. For our study, determinants were collected as quantitative clinical data from these records into a data extraction sheet by a trained research assistant. The data extraction sheet included sociodemographic, obstetric and baby related variables. The collected data was translated to English and double-entered into the electronic database EpiData 3.1. After ensuring that data from various sources were linked to the same person, the names and identifiers were removed. In Sri Lanka, pregnancy outside marriage is very rare and thus there is no place for marital status in the pregnancy record.

## Outcome measures and exposure variables

The main outcome measurement was rate of mothers with EPDS score 9 or more. Exposure variables were: maternal age (under 20, 20–29, 30–39 and 40 and more); educational level (completed secondary school ordinary level or below, completed above secondary school ordinary level); occupation (housewife or non-housewife); socioeconomic status (calculated according to the occupation of the mother and the husband, and categorized into SE 1–2 and SE 3–5); number of living children (0, 1, 2–3 and >4); former history of mental illness (yes/no); maternal Body Mass Index (BMI) (less than 18.5, 18.5–24.9, 25.0–29.9 and 30 and more); hypertension during pregnancy (yes/no); diabetes during pregnancy (yes/no); history of subfertility (yes/no); death of newborn defined as death of liveborn within 28 days (yes/no); gender of the baby (female or male); congenital abnormalities (yes/no); birthweight (less than 2500g, 2500g or more); establishment of breastfeeding (yes/no); other complications from delivery notes (yes/no).

## Data analysis

The database was exported into IBM SPSS 26 version for statistical analysis. The prevalence of mothers at risk of PPD was calculated as percentages of mothers with an EPDS score 9 or more. One-way ANOVA was performed to compare the mean EPDS score for each independent variable. Possible predictors of EPDS score 9 or more (risk of PPD) were examined in univariate and multivariable models. In the final model we adjusted with maternal age, BMI, former history of mental illness, hypertension, history of subfertility and newborn death. Odds ratios (OR) and adjusted Odds ratios (aOR) with 95% confidence intervals (95%CI) were calculated. The prevalence of mothers reporting thoughts of self-harm were calculated as percentages of mothers reporting "never", "hardly ever", "sometimes" and "many times" on the EPDS item 10; mean EPDS score was calculated for these four categories.

### Ethical issues

Ethical clearance for the present study was obtained by Ethics Review Committee, Faculty of Medicine, University of Ruhuna and by REK, Regional Committees for Medical and Health Research Ethics. Informed consent was not required as the EPDS was used as routine screening service and introduced into antenatal clinics several years before. The data archive had no names or identifiers.

## Results

Included in the study were 975 postpartum mothers. All mothers had complete EPDS records and no mothers were excluded. Among the respondents the mean age was 29 years, ranging from 16 to 45 years. The majority of the mothers (70.5%) were housewives, and half of the mothers (48.1%) had completed education above ordinary level examination secondary school. Former history of mental illness was identified in 12 women (1.2%).

Table 1 shows the EPDS scores among the 975 women studied; 9.4% (n = 92) had an EPDS score 9 or more; and 5.6% (n = 55) had a EPDS score 10 or more. An EPDS score 12 or more was seen in 2.1% (n = 20) of the participants.

### Association of postpartum depression with risk factors

Table 2 shows mean EPDS score by exposure variables and corresponding logistic regression analysis of these determinants of EPDS score 9 or more (risk of PPD). Compared to mothers aged 20–29, those aged 30–39 had twice the risk of EPDS score 9 or more (aOR 2.2, 95%CI: 1.3–3.8). Mothers who had experienced newborn death had a higher risk of EPDS score 9 or more (aOR 28.9, 95%CI: 4.5–185.1), and mothers who reported a former history of mental illness had a highly increased risk of EPDS score 9 or more (aOR 32.9, 95%CI: 7.9–136.2). Mothers with a BMI <18.5 (aOR 3.0, 95%CI: 1.4–6.3) and BMI 25.0–29.9 (aOR 2.6, 95%CI: 1.5–4.5) had a higher risk of EPDS score 9 or more than those with normal BMI. Those who had hypertension were 3.6 times more likely (aOR 3.6, 95%CI: 1.2–10.9) to develop EPDS score 9 or more as compared to mothers without hypertension.

No significant difference was found between EPDS score 9 or more and obstetric factors. Baby gender and late breast-feeding initiation was not significantly associated with the development of EPDS score 9 or more. Risk of PPD defined as EPDS score 10 or more showed similar associations with the same variables (not included in the table).

### Thoughts of self-harm

Table 3 shows responses to EPDS Item 10 about self-harm thoughts and the associated mean total EPDS score. Nearly 1 out of 5 (n = 188) study participants reported "the thought of harming myself has occurred to me". Out of those 188 participants, 26.1% had an EPDS score 9 or more. We see that the more frequent such thoughts are reported, the more likely depressive symptoms occurred.

**Table 1. Frequency distribution of the EPDS scores with sensitivity analysis among mothers in Galle, 2019.**

| EPDS | Frequency (n) | Percent (%) | 95%CI |
|---|---|---|---|
| Total | 975 | 100 | - |
| EPDS score 9 or more | 92 | 9.4 | 7.8–11.4 |
| EPDS score 10 or more | 55 | 5.6 | 4.4–7.3 |
| EPDS score 11 or more | 29 | 3.0 | 2.1–4.2 |
| EPDS score 12 or more | 20 | 2.1 | 1.3–3.1 |

**Table 2. Determinants of EPDS score 9 or more (risk of PPD) among mothers in Galle, 2019.**

| Variables | | Total | EPDS 9 or more | Mean EPDS (95% CI) | OR (95% CI) | aOR (95% CI) |
|---|---|---|---|---|---|---|
| Maternal age | | | | | | |
| | ≤19 | 52 | 2 | 4.6 (4.0–5.2) | 0.6 (0.1–2.6) | 0.8 (0.2–3.7) |
| | 20–29 | 459 | 29 | 3.8 (3.6–4.1) | 1 | 1 |
| | 30–39 | 447 | 59 | **4.6 (4.3–4.9)** | **2.3 (1.4–3.6)** | **2.2 (1.3–3.8)** |
| | ≥40 | 17 | 2 | 4.2 (2.6–5.9) | 2.0 (0.4–9.0) | 1.0 (0.1–8.6) |
| Education | | | | | | |
| | ≤Ordinary level | 505 | 46 | 4.4 (4.1–4.6) | 1 | 1 |
| | >Ordinary level | 469 | 46 | 4.1 (3.8–4.4) | 1.1 (0.7–1.7) | 1.1 (0.6–1.7) |
| | Missing | 1 | | | | |
| Occupation | | | | | | |
| | Housewife | 687 | 68 | 4.3 (4.1–4.6) | 1 | 1 |
| | Non-housewife | 288 | 24 | 4.0 (3.7–4.3) | 0.8 (0.5–1.3) | 0.7 (0.4–1.2) |
| Socioeconomic class | | | | | | |
| | 1–2 | 194 | 13 | 3.6 (3.3–4.0) | 0.6 (0.3–1.2) | 0.6 (0.3–1.2) |
| | 3–5 | 781 | 79 | 4.4 (4.2–4.6) | 1 | 1 |
| Number of living children | | | | | | |
| | 0 | 0 | 0 | – | - | - |
| | 1 | 428 | 34 | 4.1 (3.8–4.4) | 1 | 1 |
| | 2–3 | 489 | 52 | 4.3 (4.0–4.5) | 1.4 (0.9–2.2) | 0.8 (0.4–1.3) |
| | >4 | 57 | 6 | 4.5 (3.9–5.2) | 1.4 (0.5–3.4) | 0.5 (0.2–1.4) |
| | Missing | 1 | | | | |
| History of mental illness | | | | | | |
| | Yes | 12 | 9 | **11.0 (8.1–13.9)** | **31.8 (8.4–119.8)** | **32.9 (7.9–136.2)** |
| | No | 963 | 83 | 4.1 (4.0–4.3) | 1 | 1 |
| BMI | | | | | | |
| | ≤18.4 | 119 | 13 | **4.4 (3.8–5.0)** | **2.0 (1.0–4.0)** | **3.0 (1.4–6.3)** |
| | 18.5–24.9 | 462 | 27 | 4.0 (3.7–4.2) | 1 | 1 |
| | 25.0–29.9 | 290 | 43 | **4.6 (4.2–4.9)** | **2.8 (1.7–4.7)** | **2.6 (1.5–4.5)** |
| | ≥30 | 57 | 3 | 4.1 (3.4–4.9) | 0.9 (0.3–3.1) | 0.8 (0.2–2.8) |
| | Missing | 47 | | | | |
| Hypertension | | | | | | |
| | **Yes** | 17 | 6 | **6.3 (4.4–8.1)** | **5.5 (2.0–15.3)** | **3.6 (1.2–10.9)** |
| | No | 958 | 86 | 4.2 (4.0–4.4) | 1 | 1 |
| Diabetes | | | | | | |
| | Yes | 54 | 8 | 5.0 (4.0–5.9) | 1.7 (0.8–3.8) | 0.7 (0.2–1.9) |
| | No | 921 | 84 | 4.2 (4.0–4.4) | 1 | 1 |
| History of subfertility | | | | | | |
| | Yes | 39 | 7 | 5.3 (4.0–6.5) | 2.2 (0.9–5.1) | 1.0 (0.4–2.8) |
| | No | 936 | 85 | 4.2 (4.0–4.4) | 1 | 1 |
| Newborn death | | | | | | |
| | Yes | 6 | 4 | **9.0 (4.6–13.4)** | **20.0 (3.6–110.9)** | **28.9 (4.5–185.1)** |
| | No | 969 | 88 | 4.2 (4.0–4.4) | 1 | 1 |
| Baby gender | | | | | | |
| | Male | 487 | 41 | 4.2 (3.9–4.4) | 0.8 (0.5–1.2) | 0.7 (0.4–1.2) |
| | Female | 487 | 51 | 4.3 (4.0–4.6) | 1 | 1 |
| | Missing | 1 | | | | |
| Congenital abnormalities | | | | | | |

*(Continued)*

**Table 2.** (Continued)

| Variables | | Total | EPDS 9 or more | Mean EPDS (95% CI) | OR (95% CI) | aOR (95% CI) |
|---|---|---|---|---|---|---|
| | Yes | 8 | 2 | 6.0 (2.1–9.9) | 3.2 (0.6–16.3) | 1.3 (0.1–11.8) |
| | No | 966 | 90 | 4.2 (4.0–4.4) | 1 | 1 |
| | Missing | 1 | | | | |
| Birthweight | | | | | | |
| | <2500 | 89 | 6 | 4.3 (3.7–5.0) | 0.7 (0.3–1.6) | 0.5 (0.2–1.3) |
| | ≥2500 | 886 | 86 | 4.2 (4.0–4.4) | 1 | 1 |
| Breast feeding | | | | | | |
| | Yes | 429 | 35 | 4.2 (4.0–4.5) | 0.8 (0.5–1.2) | 0.6 (0.4–1.1) |
| | No | 546 | 57 | 4.2 (4.0–4.5) | 1 | 1 |
| Delivery complication | | | | | | |
| | Yes | 40 | 4 | 4.6 (3.4–5.8) | 1.1 (0.4–3.1) | 0.5 (0.2–2.0) |
| | No | 934 | 88 | 4.2 (4.0–4.4) | 1 | 1 |
| | Missing | 1 | | | | |
| Baby birth complication | | | | | | |
| | Yes | 23 | 2 | 5.0 (3.4–6.5) | 0.9 (0.2–3.9) | 0.1 (0.0–1.4) |
| | No | 936 | 90 | 4.2 (4.0–4.4) | 1 | 1 |
| | Missing | 16 | | | | |
| Malpresentation | | | | | | |
| | Yes | 45 | 1 | 4.1 (3.3–5.0) | 0.2 (0.0–1.5) | 0.1 (0.0–1.3) |
| | No | 929 | 91 | 4.2 (4.0–4.4) | 1 | 1 |
| | Missing | 1 | | | | |

Analysed by ANOVA with mean EPDS score, and univariate and multivariable logistic regressions with odds ratios. Regression analysis included these variables in the adjustment model: Maternal age, BMI, former history of mental illness, hypertension, history of subfertility and newborn death. Other variables were adjusted with the same model.

## Discussion

In this study in Sri Lanka, among 975 women screened with the EPDS four to six weeks following delivery, we found that 9.4% (95%CI: 7.8–11.4) had an EPDS score 9 or more, suggesting risk of PPD, which means they are referred to mental health services for further management and diagnosis of PPD. The factor that showed the strongest association with EPDS score 9 or more was former history of mental illness, which was not investigated separately in the former publication from 2017 [14]. Several maternal characteristics and factors were also associated with EPDS score 9 or more: high maternal age, overweight, underweight, hypertension and experiencing death of the newborn.

Various studies on PPD in Sri Lanka have shown a wide variation in prevalence, perhaps partly due to differences in methodology. In 2017, a similar study conducted in Galle, Sri

**Table 3. EPDS question 10: "The thought of harming myself has occurred to me".**

| Response | Frequency (n) | Percent (%) | Mean total EPDS (95%CI) |
|---|---|---|---|
| Total | 975 | 100 | - |
| Never | 787 | 80.7 | 3.6 (3.4–3.8) |
| Hardly ever | 174 | 17.8 | 6.6 (6.2–7.0) |
| Sometimes | 14 | 1.4 | 8.3 (7.0–9.6) |
| Many times | 0 | 0 | - |

Lanka reported that 7.8% mothers had an EPDS score 10 or more [14]; our study showed 5.6% (95%CI: 4.4–7.3) with this cut-off. Our data support the previous finding of low risk of PPD in Galle, Sri Lanka. However, we see that when measuring the prevalence rates of mothers at risk of PPD using a EPDS cut-off 9 or more compared to a cut-off 10 or more, a difference of 3.8% in prevalence estimates occurred. This supports the statement by Matthey et al. that a one point difference in EPDS cut-off score can have a substantial impact on PPD rate estimates and highlights the importance of using the validated cut-off for screening in Sri Lanka [8]. In human development report for Sri Lanka in 2012, Galle ranked between mid and top on health index and high on education index [17]; this may contribute to a low prevalence of PPD. Furthermore, the prevalence of mothers at risk of PPD is lower than findings in other countries in South-Asia. A validation study from Nepal showed a prevalence of PPD at 17.1%. It defined PPD as EPDS score 13 or more and had a sensitivity of 92% and a specificity of 95.6% [18].' A meta-analysis from India estimated a pooled prevalence of PPD at 22% [19]. Compared to the neighboring countries, Sri Lanka has made more progress on maternal health and child mortality indicators, as well as other indicators such as GDP per capita and education [20].

In our study, mothers aged 30–39 had twice the risk of EPDS score 9 or more compared to those aged 20–29. The same findings are reported in the paper from the same place in 2017 [14]. A higher percentage of EPDS score 9 or more in the age group 30–39 compared to other age groups was observed in both primiparous mothers and in multiparous mothers. Other sociodemographic risk factors such as educational level and being a housewife did not show significant associations with EPDS score 9 or more.

In terms of maternal BMI, underweight women (BMI<18.5) had a three times higher risk of EPDS score 9 or more compared to women of normal weight, which is consistent with findings in Nepal [21] and Sweden [22]. It is widely recognized that BMI is likely to be influenced by income [17] and after adjusting for only socioeconomic class, the association between underweight and EPDS score 9 or more was no longer significant. In Sri Lanka, overweight (BMI 25.0–29.9) is far more prevalent than underweight and was in our study associated with more than twice the risk of EPDS score 9 or more compared to mothers with normal bodyweight. One study suggests that overweight in Sri Lanka is associated with urban living and being in middle age [23].

We found that hypertension was associated with increased risk of EPDS score 9 or more. In our study, the mothers with hypertension were more often of advanced age, which reflects the prevalence of hypertension in the general population [24]. Also, hypertensive mothers had more gestational diabetes, also increasing with age. The possible relationship between pre-eclampsia and PPD has previously been described in studies from South Korea [25] and Netherlands [26]. In pre-eclampsia, the blood level of serotonin is known to increase [27] and previous studies have hypothesized that this may be linked to a decreased level of serotonin in the brain [26]. A possible role of alterations in serotonin function has been noted in the pathophysiology of general depression [28]. A further complication may be that hypertension in pregnancy is commonly treated with methyldopa, of which a known side effect is depressive symptoms [29]. However, hypertension and preeclampsia could not be differentiated from the pregnancy record in our data.

In our study mothers who had experienced death of a newborn were at increased risk of EPDS score 9 or more, although grief reactions may be difficult to distinguish from PPD symptoms. We also found that former history of mental illness predicts EPDS score 9 or more, which is consistent with the findings by O'Hara and Swain [4]. In our study, a diagnosis of mental illness before the pregnancy was included but not further defined. However, the number of mothers identified (n = 12) may not represent the total number of mothers who ever had a mental illness; perhaps those with a more recent diagnosis were more likely to report on

this item but we do not have this information. In Sri Lanka, stigmatizing beliefs on mental health have been reported to affect help-seeking behavior [30]. As in many Asian cultures, emotional restraint and self-control are perceived as desirable qualities in women, and to seek counselling for mental health issues may be considered unacceptable by family members [30]. Our findings suggest that attendants reporting a past mental illness may be considered at risk of PPD, and may benefit from referral.

Thoughts of self-harm was reported by almost 20%, of whom the vast majority responded "hardly ever". In a randomized controlled trial in the UK, it was found that the response "hardly ever" on EPDS item 10 about thoughts of self-harm in mothers was not concordant with suicidality as measured by a clinical interview [31]. A study published in 2019 which included 475 pregnant women in Anuradhapura, Sri Lanka reported rates on thoughts of self-harm at 5.9% including "hardly ever" [32], and in Goa, India, it was 14.3% [9, 33]. In Pittsburgh, USA, with a study population of 10 000 mothers, the observed rate on thoughts of self-harm four to six weeks postpartum was 3.2%, out of whom the majority had screen-positive findings on depression [34]. This is in contrast to our study, where only one out of four women with thoughts of self-harm also had an EPDS score 9 or more; however, we noticed that the more frequent the mothers' self-harm thoughts were, the more signs of depression occurred, shown by increasing mean EPDS score. In Sri Lanka, mothers are usually not referred to a psychiatrist based on the response on EPDS item 10. However, research in Sri Lanka on self-harm behavior following delivery is limited. The high proportion of mother with thoughts of self-harm in our study highlights the need for more research in this field.

Our study had several strengths. The study population provides some confidence to the precision in our results, and missing data was minimal. The transfer of information in paper forms into digital data was validated by double data entry. We think the study gives a good representation of the situation for pregnant women in the area, as almost all pregnant women there attend ANC. Furthermore, our study is the first study since the EPDS was included as a part of postnatal care in 2012 to use the validated cut-off for screening when estimating mothers at risk of PPD in Sri Lanka.

The study also had some limitations. The cross-sectional study design can only analyze associations and not causation, but this is in line with our aims. Recall is generally associated with some uncertainty, especially about feelings in the past, but limiting to the past week may provide more credibility to the findings. Misinterpretation of the questions and social taboos may have led to response bias, but the local author did not see such obvious mistakes. The cut-off threshold for categorizing depression is a matter of discussion; we used a validated cut-off for screening for mothers at risk of PPD in Sri Lanka, and we also tested the association with other cut-offs as well (Table 1) and found similar associations (not shown in table). The tool we used was tested and validated in both Sinhala and Tamil language. Nonetheless, great caution must be taken when comparing the results of EPDS screening for PPD in different countries as there is no international standard for how to translate, nor validate, the EPDS in local languages [35].

## Conclusion

The prevalence of mothers at risk of PPD in Galle, Sri Lanka (as estimated by an EPDS score 9 or more) was 9.4% (95%CI: 7.8–11.4). The prevalence rate was lower when using higher cut-offs for screening which highlights the importance of using the validated cut-off EPDS score 9 or more for future studies on PPD in Sri Lanka. The major predisposing factors for EPDS score 9 or more (risk of PPD) were former history of mental illness, high maternal age, overweight, hypertension and newborn death. Furthermore, the number of mothers reporting

thoughts of self-harm was high. Mothers at increased risk of PPD are important targets of referral to specialist.

## Supporting information

**S1 Dataset.**
(SAV)

## Acknowledgments

The authors would like to thank all Public Health midwives of Bope-Poddala MOH division for their effort and cooperation in collecting data for this study, as well as our research assistant Ms. Prabodha Lakmali. We would like to thank Prof. Ganesh Acharya for his insightful suggestions and reading of the manuscript. This research was supported by University of Ruhuna, Sri Lanka and University of Bergen, Norway.

## Author Contributions

**Conceptualization:** Therese Røysted-Solås, Sven Gudmund Hinderaker, Lasantha Ubesekara, Vijitha De Silva.

**Data curation:** Vijitha De Silva.

**Formal analysis:** Therese Røysted-Solås, Sven Gudmund Hinderaker.

**Investigation:** Therese Røysted-Solås.

**Methodology:** Therese Røysted-Solås, Sven Gudmund Hinderaker, Vijitha De Silva.

**Project administration:** Therese Røysted-Solås, Vijitha De Silva.

**Supervision:** Sven Gudmund Hinderaker.

**Writing – original draft:** Therese Røysted-Solås.

**Writing – review & editing:** Sven Gudmund Hinderaker, Lasantha Ubesekara, Vijitha De Silva.

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
