## [Decision Letter · Decision Letter 0]

6 Sep 2021

PONE-D-21-20712Postpartum depression and associated risk factors among mothers in Galle, Sri LankaPLOS ONE

Dear Dr. Therese Roysted Solas,

Thank you for submitting your manuscript to PLOS ONE. After careful consideration, we feel that it has merit but does not fully meet PLOS ONE’s publication criteria as it currently stands. Therefore, we invite you to submit a revised version of the manuscript that addresses the points raised during the review process.

ACADEMIC EDITOR:

Please address the comments mentioned in the additional comments section below:

We look forward to receiving your revised manuscript.

Kind regards,

Yeetey Akpe Kwesi Enuameh, MD, MSc, DrPH

Academic Editor

PLOS ONE

Journal Requirements:

Additional Editor Comments (if provided):

The topic is of great relevance to maternal and child health... there are some issues that when addressed should enhance the outlook of the paper... Please address the comments of the reviewer...

1. The title of the current paper and that by Fan et al., 2020 are quite similar. Could you please come out clearly as to what differs between your paper and the earlier published one, more so when the study population seems to be the same.

2. Lines 17 & 40: The prevalence estimates provided here were 1996 figures... could there not have been much more recent ones? Also add the estimates from South Asia - as you compare it in the conclusion...

3. Lines 56 - 58: I am not clear about this point... what has this to do with PPD? I read over the cited article, did not come across a relationship between poverty and quality of parenting... please clarify...

4. Lines 92 - 93: Why the assumption based on 1996 figures? More so, when there was a 2017 study in the same population?

5. Lines 94 - 95: Why all mothers????? We the readers are not privy to information on deliveries in the study community... Also, the results section (line 159) has 975 persons... you have not clearly stated how was that arrived at... were those all the mothers interviewed? any mothers not included? response rate?

6. Line 98: Will suggest this section be structured as e.g., data collection methods, tools, and process.

7. Lines 106 - 107: check this.... specificity????

8. Lines 109 - 115: Is the information presented here related to data collection?

9. Line 114: Is the ICD used for classification or diagnosis?

10. Line 126: Label this as a sub-section e.g., variables considered.

11. Lines 159 - 162: Please make reference to table 2.

12. Lines 193 - 195: Revise the title to be much more concise.

13. Line 201: Mention "EPDS Question 10" in the "Methods section" as part of the data collection tools.

14. Lines 214 - 216: Were these characteristics also in the 2017 study? If so, state that clearly... if not, then provide the appropriate citation.

15. Finally, please follow the reporting guidelines for Cross-sectional studies as provided in the link to guide your paper: https://www.equator-network.org/reporting-guidelines/strobe/. Also review the PLOS ONE guidelines on publications: https://journals.plos.org/plosone/s/criteria-for-publication

Reviewers' comments:

Reviewer's Responses to Questions

**Comments to the Author**

1. Is the manuscript technically sound, and do the data support the conclusions?

Reviewer #1: Partly

2. Has the statistical analysis been performed appropriately and rigorously? 

Reviewer #1: Yes

3. Have the authors made all data underlying the findings in their manuscript fully available?

Reviewer #1: Yes

4. Is the manuscript presented in an intelligible fashion and written in standard English?

Reviewer #1: Yes

5. Review Comments to the Author

Reviewer #1: The study addresses a key issue however there are some modifications required.

Background

The authors provide some useful information about postpartum depression in the background. Considering that a very similar study had already been conducted in the catchment area however, I think the authors could have provided more information on PPD in Sri Lanka and Galle. In addition, information on the validated tools used in measuring PPD in Sri Lanka as stated in the previous study are also relevant to this study. They could have been stated and appropriately referenced in this section as well.

For example, the fact that the EPDs have been translated into Sinhalese and validated in Sri Lanka is important information that should be made known in the background paper. Additionally, PPD in Sri Lanka is reported to have decreased since 2011 and the background of this paper should have captured this information.

Justification of study: Lines 71-74

The study was justified on the grounds that the similar study conducted in the same area in 2017 showed a PPD lower than that of another study conducted in Sri Lanka in 2004. However, a lot could have happened between 2004 to 2017 when the new study was conducted. Various reasons could have contributed to the difference and a proper review of the programs and interventions put in place between 2004 to 2017 could have provided more information as to why this difference was observed. Additionally, the 2017 paper indicated that there had been a general reduction in PPD from 2011, so the reduction should not really come as a surprise anyway. In my view therefore, there is not a strong justification for this study and it does not really offer anything new as it stands.

Recommendation: A comparative analysis or trend analysis between the previous studies from 2011 to 2017 will give a better justification for the study and provide a clearer picture of PPD in Sri Lanka. Therefore, if the data is available, I recommend that the authors provide some more information on PPD between 2011 to 2017. The authors can also make a stronger justification for the study using the timing of their administration of their EPDs if it differs significantly from other studies in Sri Lanka

Methods

The collection of data using the EPD form is not really clear. Was the PPD information collected as part of the routine data collection or the team specifically went out to the field and collected this data using midwives?

This distinction should be made clearer in the write up. It is clear that the team made use of routine records for the other variables but as things stand now one gets the impression that the PPD information was collected in the same way. I am not sure that is the case however if the previous 2017 study is anything to go by.

Additionally, if this is not the case then, I am also wondering why informed consent (verbal at least) was not required? Was this study simply a secondary analysis of routine data?

How often is screening done for PPD at the facility? Was the EPD form only employed because of this current study? If the screening form and PPD data is readily available, then the team could also do a trend analysis or provide some more information on PPD in the catchment area prior to the study.

Lines 104-109: The study made use of the Tamil and Sinhala versions of the EPDS. Additionally, the study also talks about illiterate participants. Who are the illiterate participants? Are illiterate participants in this case participants who could not read Tamil and Sinhala?

Results

The results are well presented. Some of the confidence intervals are really wide though and the authors should take a second look at their analyses and the assumptions made for their models. Example New born death aOR 28.9 (4.5-185.1). The authors did not present any findings from their chi square analysis.

Discussion

Line 209: According to the authors they dealt with women who had just given birth. However, in earlier chapters, they spoke about women who had given birth 4 to 6 weeks ago. The authors should be consistent as time is important in measurement of PPD. The study in 2017 for example had two time points (10 days and 4 weeks).

Recommendation: In the 2017 study, it was recommended that further studies on the effect of time since delivery on PPD should be looked at. Since the form is administered routinely, the authors can look at the different time periods that the forms are administered to make a stronger case about PPD in Sri Lanka.

Conclusion

The conclusion here is much better than that the one in the abstract. The authors could look at revising the one in the abstract

6. PLOS authors have the option to publish the peer review history of their article (what does this mean?). If published, this will include your full peer review and any attached files.

Reviewer #1: No

---

## [Author Response · Author response to Decision Letter 0]

6 Oct 2021

EDITOR

Additional Editor Comments:

Comment 1: The title of the current paper and that by Fan et al., 2020 are quite similar. Could you please come out clearly as to what differs between your paper and the earlier published one, more so when the study population seems to be the same 

Response: Thank you for pointing this out. In the revised version we have chosen to highlight that this study uses the validated cut-off for screening for PPD. The title is changed to “Risk of postpartum depression in Sri Lanka: A population-based study using a validated screening tool”.

Comment 2: Lines 17 & 40: The prevalence estimates provided here were 1996 figures... could there not have been much more recent ones? Also add the estimates from South Asia - as you compare it in the conclusion 

Response: Good point. We have changed the reference on prevalence estimates in line 16 and line 104 to a more recent one by Hahn-Holbrook (2018). We also added prevalence ranges of PPD in South Asia in the introduction in line 73. However, we removed the sentence “this is lower than other countries in South Asia” in the conclusion as we have put more focus to screening for PPD in Sri Lanka in the revised version.

Comment 3: Lines 56 - 58: I am not clear about this point... what has this to do with PPD? I read over the cited article, did not come across a relationship between poverty and quality of parenting... please clarify...

Response: We have deleted this sentence.

Comment 4: Lines 92 - 93: Why the assumption based on 1996 figures? More so, when there was a 2017 study in the same population?

Response: At the time of planning and estimating our sample size we did not have the 2017 figures. As we knew that around 1000 births take place each calendar year in Galle, we estimated that in our sample 1000 mothers would give a reasonable precision and power. We calculated samples size both using the 1996 figures and using lower hypothetical figures with not related references. Basically our considerations tested more but only referred to the published old number. As we have now changed the 1996 reference in the introduction to a more recent one, we also updated the sample size estimation reference to be 18% Hahn-Holbrook (2018).

Comment 5: Lines 94 - 95: Why all mothers????? We the readers are not privy to information on deliveries in the study community... Also, the results section (line 159) has 975 persons... you have not clearly stated how was that arrived at... were those all the mothers interviewed? any mothers not included? response rate?

Response: Thank you for pointing this out. Basically all women who attended clinic after delivery are asked to fill the EPDS questionnaire screening them for postpartum depression. It is part of the routine public services at Galle, and we do not know whether any of them refused to fill the form, as we have only the filled forms. Exclusion criteria was incomplete EPDS records. We adjusted the methods section according to you recommendation, and added a line in the first paragraph in results section in line 177 to make this clearer.

Comment 6: Line 98: Will suggest this section be structured as e.g., data collection methods, tools, and process.

Response: Thank you for this great suggestion. We have now made the following subsections under methods “study design and study setting”, “participants”, “study tool”, “data collection”, “outcome measures and exposure variables”, “data analysis” and “ethical issues”. 

Comment 7: Lines 106 - 107: check this.... specificity????

Response: Thank you for notifying this mistake. It has now been corrected in line 117-118.

Comment 8: Lines 109 - 115: Is the information presented here related to data collection?

Response: No, the information presented was related to the current practice on screening for PPD in Sri Lanka. Thank you for notifying this. We have now divided the methods section into more subheadings to separate our data collection from the study tool according to your recommendation.

Comment 9: Line 114: Is the ICD used for classification or diagnosis?

Response: Yes, we have now changed the line 127 to be more precise. 

Comment 10: Line 126: Label this as a sub-section e.g., variables considered.

Response: Thank you for this recommendation, we agree to add a subsection before this paragraph and have adjusted it accordingly.

Comment 11: Lines 159 - 162: Please make reference to table 2.

Response: Table 2 is now referred to in the text line 178. 

Comment 12: Lines 193 - 195: Revise the title to be much more concise.

Response: We have shortened the title of table 2.

Comment 13: Line 201: Mention "EPDS Question 10" in the "Methods section" as part of the data collection tools.

Response: Thank you for this recommendation. We added this information under methods section in line 118.

Comment 14: Lines 214 - 216: Were these characteristics also in the 2017 study? If so, state that clearly... if not, then provide the appropriate citation.

Response: Thank you, this needed clarification. The factors in our study that showed the strongest association with PPD was former history of mental illness and death of the newborn, both not mentioned in the 2017 study. Hypertension was associated with PPD and not studied separately in the former study. High maternal age was associated with PPD in the former study, and this is mentioned in maternal age section in discussion. 

Comment 15: Finally, please follow the reporting guidelines for Cross-sectional studies as provided in the link to guide your paper: https://www.equator-network.org/reporting-guidelines/strobe/. Also review the PLOS ONE guidelines on publications: https://journals.plos.org/plosone/s/criteria-for-publication

Response: Thank you for providing us with this information. We will adjust the paper according to the guidelines provided. 

REVIEWER

Reviewer #1: The study addresses a key issue however there are some modifications required.

Background

Comment 16: The authors provide some useful information about postpartum depression in the background. Considering that a very similar study had already been conducted in the catchment area however, I think the authors could have provided more information on PPD in Sri Lanka and Galle. In addition, information on the validated tools used in measuring PPD in Sri Lanka as stated in the previous study are also relevant to this study. They could have been stated and appropriately referenced in this section as well.

For example, the fact that the EPDS have been translated into Sinhalese and validated in Sri Lanka is important information that should be made known in the background paper. Additionally, PPD in Sri Lanka is reported to have decreased since 2011 and the background of this paper should have captured this information.

Response: Thank you for these great suggestions. We agree that the findings from 2011 should have been included in the introduction and we have adjusted it accordingly. In the introduction we also added new information about use of cut-off scores when screening for PPD, as well as information the validated cut-off for screening in Sri Lanka and current practice. 

Comment 17: Justification of study: Lines 71-74

The study was justified on the grounds that the similar study conducted in the same area in 2017 showed a PPD lower than that of another study conducted in Sri Lanka in 2004. However, a lot could have happened between 2004 to 2017 when the new study was conducted. Various reasons could have contributed to the difference and a proper review of the programs and interventions put in place between 2004 to 2017 could have provided more information as to why this difference was observed. Additionally, the 2017 paper indicated that there had been a general reduction in PPD from 2011, so the reduction should not really come as a surprise anyway. In my view therefore, there is not a strong justification for this study and it does not really offer anything new as it stands.

Response: We thank the reviewer for the very interesting comment. We would like to mention that the study from 2017 was published after we had collected our data in 2019 and the results were not available at the time when we made the research protocol. From your input we now realize that we should have been clearer on this issue, and not use the paper from 2017 as an argument for conducting a similar study over again as this study was not available at the time we gathered data. Nevertheless, we believe that our study provide new information about PPD in Sri Lanka as our study is the first study to use the validated cut-off 9 or more for screening for PPD in Sri Lanka. Previous studies on PPD in Sri Lanka have used the cut-off 10 or more for reasons we are not aware of. However, the validated cut-off for screening is 9 or more and we believe that our study provides more credibility to the findings on prevalence on PPD in Sri Lanka. Furthermore, our study focuses on individual risk factors including diseases during pregnancy as well as former history of mental illness, of which the 2017 study did not provide information. For future studies on postpartum and antenatal depression in Sri Lanka it is important to use the validated cut-off for screening for postnatal depression.

Comment 18: Recommendation: A comparative analysis or trend analysis between the previous studies from 2011 to 2017 will give a better justification for the study and provide a clearer picture of PPD in Sri Lanka. Therefore, if the data is available, I recommend that the authors provide some more information on PPD between 2011 to 2017. The authors can also make a stronger justification for the study using the timing of their administration of their EPDs if it differs significantly from other studies in Sri Lanka

Response: Thank you for good suggestions. In our study, the mothers are screened once at clinics for PPD between 4-6 weeks following delivery, not at two different timepoints. The 2017 study screened once for postpartum depression at 10 days following delivery in Dankotuwa district and once in Galle district 4 weeks following delivery. As two different times for screening was not completed within the same population, we believe that there are several sociodemographic confounding factors that could have contributed to these results. Furthermore, when testing after 10 days the findings of PPD prevalence may be confounded by the common “postpartum blues” which is limited to 2 weeks following delivery. We think by using 4 weeks instead of the first 10 days postpartum we give a better picture of the real depressive postpartum conditions that appear. 

Methods

Comment 19: The collection of data using the EPD form is not really clear. Was the PPD information collected as part of the routine data collection or the team specifically went out to the field and collected this data using midwives?

This distinction should be made clearer in the write up. It is clear that the team made use of routine records for the other variables but as things stand now one gets the impression that the PPD information was collected in the same way. I am not sure that is the case however if the previous 2017 study is anything to go by

Response: Thank you for pointing this out. We agree and have adjusted the text to be clearer. 

Comment 20: Additionally, if this is not the case then, I am also wondering why informed consent (verbal at least) was not required? Was this study simply a secondary analysis of routine data?

Response: The EPDS and pregnancy records are routinely administered in all mothers in Sri Lanka, meant for screening and potentially referral for further assessment and management. Therefore, as you have stated, the study is a secondary analysis of routine data. However, we understand that we were not clear on this and have adjusted the manuscript accordingly.

Comment 21: How often is screening done for PPD at the facility? 

Response: The screening is completed once in Galle, Sri Lanka, and it is administered between 4-6 following delivery. This study digitalized the routine copies of the forms to analyze the data that is stored in the records of the clinic clients. 

Comment 22: Was the EPD form only employed because of this current study? If the screening form and PPD data is readily available, then the team could also do a trend analysis or provide some more information on PPD in the catchment area prior to the study.

Response: Thank you for good suggestions. The EPDS is a part of a routine screening service. However, we are not in the possession of PPD data from years before, and accessing this data is not possible. Therefore, a trend analysis is unfortunately not possible to conduct. 

Comment 23: Lines 104-109: The study made use of the Tamil and Sinhala versions of the EPDS. Additionally, the study also talks about illiterate participants. Who are the illiterate participants? Are illiterate participants in this case participants who could not read Tamil and Sinhala?

Response: Thank you for pointing this out. Agree, we have accordingly revised line 121-122 to emphasize who the illiterate participants are. 

Results

Comment 24: The results are well presented. Some of the confidence intervals are really wide though and the authors should take a second look at their analyses and the assumptions made for their models. Example New born death aOR 28.9 (4.5-185.1). The authors did not present any findings from their chi square analysis.

Response: Thank you. We agree with this comment. As you know when the cases are few the confidence intervals of the estimates are wide, which is what we see with the mothers who lost their child. We prefer to give estimates and statistical significance shown as 95% confidence limits instead of p-values; however, if the editor insists, we can easily provide p-values in addition to the 95% CI.

Discussion

Comment 25: Line 209: According to the authors they dealt with women who had just given birth. However, in earlier chapters, they spoke about women who had given birth 4 to 6 weeks ago. The authors should be consistent as time is important in measurement of PPD. The study in 2017 for example had two time points (10 days and 4 weeks). Recommendation: In the 2017 study, it was recommended that further studies on the effect of time since delivery on PPD should be looked at. Since the form is administered routinely, the authors can look at the different time periods that the forms are administered to make a stronger case about PPD in Sri Lanka.

Response: We agree and we have adjusted the text to be clearer in line 230. Regarding effect of time since delivery, please see the response to comment 18 which explains why a study on different times for screening is not possible to conduct with the data we have available. 

Conclusion

Comment 26: The conclusion here is much better than that the one in the abstract. The authors could look at revising the one in the abstract

Response: Thank you. We have adjusted the conclusion in the abstract accordingly.

---

## [Decision Letter · Decision Letter 1]

22 Feb 2022

PONE-D-21-20712R1Risk of postpartum depression in Sri Lanka: A population-based study using a validated screening toolPLOS ONE

Dear Dr. Solas,

Thank you for submitting your manuscript to PLOS ONE. After careful consideration, we feel that it has merit but does not fully meet PLOS ONE’s publication criteria as it currently stands. Therefore, we invite you to submit a revised version of the manuscript that addresses the points raised during the review process.

Some issues related to consistency of expressions used, grammatical and structural errors to not make the manuscript fit for publication at this point in time. Please patiently and meticulously address the issues raised below and resubmit.

We look forward to receiving your revised manuscript.

Kind regards,

Yeetey Akpe Kwesi Enuameh, MD, MSc, DrPH

Academic Editor

PLOS ONE

Journal Requirements:

Additional Editor Comments:

Thanks to the authors for the efforts at revising the manuscript. Though Reviewer 1 has cleared the paper for publication, there are still some issues to be addressed to enhance its outlook. I have made comments on the part of the paper with tracked changes. There are several grammatical and structural errors in the manuscript, as such I would recommend it being submitted for proof-reading to rectify most of those errors.

1. Though the title has been revised, the content of the paper seems to be at variance. Is the focus on "Risk of PPD", OR "Prevalence of PPD" or "Prevalence of risk of PPD"? That should come out clearly and consistently throughout the manuscript.

ABSTRACT

2. Lines 40 & 41: Please revise the grammar and sentence structure to enhance clarity.

3. Line 42: Should the sentence not be "The prevalence of PPD among mothers with EPDS score 9 and above was..."? To the best of my understanding, the prevalence is of PPD and not EPDS.

4. Line 47: The "thoughts of self-harm" should be appropriately linked to PPD... Just mentioning it does not add any value to its presence...

5. Line 48: can 9.4% be referred to as "a significant proportion"???

6. Lines 50 & 51: Would it not have been simpler to just say as the data shows that "a higher cut-off produces reduced PPD prevalence estimates"? The current statement is quite ambiguous...

INTRODUCTION

7. Line 64: "suggest" or "suggests"?

8. Lines 73 & 74: how is this related to PPD? that would be very helpful to contextualize the information on "self-harm".

9. Line 94: "put focus" does not seem to be the right expression

10. Line 98, 104 & 105: Revise the sentences grammatically and structurally to enhance its clarity

11. Lines 106: Is the focus of the study "Prevalence of mothers at risk of PPD" or "prevalence of mothers with PPD"????

12. Line 113: Under point 1, which study is being "repeated"?

13. Line 114: Postpartum women do not attend "antenatal clinics", so please revise

PARTICIPANTS

14. Line 138: Would it not be best to add that this sample size (907) was close to the 975 that delivered over the period and as such they were all included in the study?

15. Line 140 & 141: Why not say "mothers with incomplete EPDS records were excluded from the study"?

STUDY TOOL

16. Line 152 & 153: The sentence seems to be contained in the next, so why not incorporate one into the other?

17. Line 156: Would be great if the influence of "thoughts of self-harm" on PPD could be clearly stated

DATA COLLECTION

18. Line 170: pregnant women in this study cannot have EPDS records as the information is collected after 4 weeks of delivery..

DATA ANALYSIS

19. Lines 199 - 202: The non-use of the Pearson Chi-square test in data analysis was raised by Reviewer 1.... this was not reflected in the results section...

20. Lines 206 - 208: "Thoughts of self-harm" seem to be a very important feature of this study - could you clearly state the importance of this to PPD or its influence on the study outcomes clearly?

RESULTS

21. Line 218: The study was not about "pregnant women", so please revise

DISCUSSION

22. Line 276: which is the former publication??? any citations?

23. Line 281: "Newborn death" is not a maternal characteristic

24. Line 286: What are you implying with the statement "our material"???

25. Line 293: The statement is contrary to that of the first sentence in the conclusion of the abstract...

26. Line 299: Check your grammar

27. Line 360: the available "study population" was used... why not state that instead of "adequate sample size"?

28. Line 366: is it "prevalence of PPD" or "risk of PPD" or "prevalence of risk of PPD"???? Please be consistent all through the manuscript...

CONCLUSION

29. Line 381: Should the initial part of the sentence not be "the prevalence of PPD" or "risk of PPD at an EPDS score of 9 or more"???? Thought it was the "prevalence of PPD being measured with the EPDS" and "not the prevalence of EPDS".

30. Line 384: In place of "had a substantial impact", why not state clearly that "there was a drop in prevalence rates with higher cut-offs for EPDS"??? That would be much more to the point

Reviewers' comments:

Reviewer's Responses to Questions

**Comments to the Author**

1. If the authors have adequately addressed your comments raised in a previous round of review and you feel that this manuscript is now acceptable for publication, you may indicate that here to bypass the “Comments to the Author” section, enter your conflict of interest statement in the “Confidential to Editor” section, and submit your "Accept" recommendation.

Reviewer #1: All comments have been addressed

2. Is the manuscript technically sound, and do the data support the conclusions?

Reviewer #1: Yes

3. Has the statistical analysis been performed appropriately and rigorously? 

Reviewer #1: Yes

4. Have the authors made all data underlying the findings in their manuscript fully available?

Reviewer #1: Yes

5. Is the manuscript presented in an intelligible fashion and written in standard English?

Reviewer #1: Yes

6. Review Comments to the Author

Reviewer #1: (No Response)

7. PLOS authors have the option to publish the peer review history of their article (what does this mean?). If published, this will include your full peer review and any attached files.

Reviewer #1: No

---

## [Author Response · Author response to Decision Letter 1]

22 Mar 2022

General comment

We have now corrected grammatical errors and sentence structure in numerous sections. Furthermore, we added a paragraph in the introduction about thoughts of self-harm, and moved the self-harm paragraph in the discussion to be after the paragraph about newborn death and former history of mental illness as risk factors for EPDS score 9 or more (risk of PPD). We also added a new reference to the reference list in line 441.

Additional Editor Comments

Comment 1: Though the title has been revised, the content of the paper seems to be at variance. Is the focus on "Risk of PPD", OR "Prevalence of PPD" or "Prevalence of risk of PPD"? That should come out clearly and consistently throughout the manuscript.

Response: Thank you for this comment. Our focus is the prevalence of mothers at risk of PPD (as measured by mothers with an EPDS score 9 or more). We have now edited the manuscript and title to be consistent in terminology. 

ABSTRACT

Comment 2: Lines 40 & 41: Please revise the grammar and sentence structure to enhance clarity.

Response: Thank you for pointing this out, we have changed the sentence in line 22-24 to make it clearer.

Comment 3: Line 42: Should the sentence not be "The prevalence of PPD among mothers with EPDS score 9 and above was..."? To the best of my understanding, the prevalence is of PPD and not EPDS.

Response: Thank you for raising this issue. The EPDS is a screening tool to identify mothers at risk of PPD, but a final diagnosis is made through clinical interview with the mothers identified. Therefore, we believe that the sentence “mothers with an EPDS score 9 or more (risk of PPD)” is more precise than stating “mothers with PPD”; and we have adjusted the terminology throughout the manuscript to be consistent on this matter; in the risk factor sections we have changed the wording to “risk of EPDS score 9 or more”. As previous studies in Sri Lanka have used a higher cut-off for screening for PPD, the prevalence estimates cannot be compared correctly without stating which EPDS score is being used. 

However, we can use the terminology “prevalence of PPD” consistently throughout the paper if Editor prefers, but we believe that this terminology is less precise. 

Comment 4: Line 47: The "thoughts of self-harm" should be appropriately linked to PPD... Just mentioning it does not add any value to its presence...

Response: Thank you, this needed clarification. We have added a new paragraph in the introduction, to describe the relationship between serf-harm and PPD in line 86-92.

Comment 5: Line 48: can 9.4% be referred to as "a significant proportion"???

Response: We have adjusted the line 39.

Comment 6: Lines 50 & 51: Would it not have been simpler to just say as the data shows that "a higher cut-off produces reduced PPD prevalence estimates"? The current statement is quite ambiguous...

Response: We agree to this comment, and have adjusted the line 41-44.

INTRODUCTION

Comment 7: Line 64: "suggest" or "suggests"?

Response: We have corrected the line 56.

Comment 8: Lines 73 & 74: how is this related to PPD? that would be very helpful to contextualize the information on "self-harm".

Response: Thank you for pointing this out. We have added a paragraph in line 86-92 to make this more clear.

Comment 9: Line 94: "put focus" does not seem to be the right expression

Response: We agree to this point and have adjusted the sentence in line 80.

Comment 10: Line 98, 104 & 105: Revise the sentences grammatically and structurally to enhance its clarity

Response: The sentences has been adjusted in line 95 and line 102-103.

Comment 11: Lines 106: Is the focus of the study "Prevalence of mothers at risk of PPD" or "prevalence of mothers with PPD"????

Response: Thank you for addressing the inconsistent use of terminology. As stated in comment 3, the focus in our paper is mothers at risk of PPD defined as mothers with an EPDS score 9 or more and we have adjusted the manuscript and title to be consistent.

Comment 12: Line 113: Under point 1, which study is being "repeated"?

Response: We have changed the sentence in line 107-108.

Comment 13: Line 114: Postpartum women do not attend "antenatal clinics", so please revise

Response: We have changed the line 107-108.

PARTICIPANTS

Comment 14: Line 138: Would it not be best to add that this sample size (907) was close to the 975 that delivered over the period and as such they were all included in the study?

Response: Thank you for this comment. However, as the sample size estimation was completed before we had the actual number of births in 2019, we believe that our sentence describes this more clearly.

Comment 15: Line 140 & 141: Why not say "mothers with incomplete EPDS records were excluded from the study"?

Response: We have adjusted the sentence in line 133-134 accordingly.

STUDY TOOL

Comment 16: Line 152 & 153: The sentence seems to be contained in the next, so why not incorporate one into the other?

Response: We incorporated the sentences in line 141-144. 

Comment 17: Line 156: Would be great if the influence of "thoughts of self-harm" on PPD could be clearly stated

Response: We added a paragraph in the introduction to describe the relationship between PPD and self-harm. We deleted the sentence in line 145. 

DATA COLLECTION

Comment 18: Line 170: pregnant women in this study cannot have EPDS records as the information is collected after 4 weeks of delivery.

Response: Good point, we have changed the line 158 to be more precise. 

DATA ANALYSIS

Comment 19: Lines 199 - 202: The non-use of the Pearson Chi-square test in data analysis was raised by Reviewer 1.... this was not reflected in the results section...

Response: As we have not used the results of the Pearson Chi-square in the tables, we have now deleted the sentence in line 186-188. 

Comment 20: Lines 206 - 208: "Thoughts of self-harm" seem to be a very important feature of this study - could you clearly state the importance of this to PPD or its influence on the study outcomes clearly?

Response: We have inserted a paragraph in the introduction to describe the importance of self-harm thoughts in PPD and have adjusted the discussion in line 353-355 to be more precise.

RESULTS

Comment 21: Line 218: The study was not about "pregnant women", so please revise

Response: We have adjusted the line 205. 

DISCUSSION

Comment 22: Line 276: which is the former publication??? any citations?

Response: Citation has been added in line 270.

Comment 23: Line 281: "Newborn death" is not a maternal characteristic

Response: We agree to this comment and have adjusted the line 272-273. 

Comment 24: Line 286: What are you implying with the statement "our material"???

Response: We have changed the wording in line 276-278. 

Comment 25: Line 293: The statement is contrary to that of the first sentence in the conclusion of the abstract...

Response: The conclusion has been changed in line 39-40.

Comment 26: Line 299: Check your grammar

Response: It has been corrected in line 292.

Comment 27: Line 360: the available "study population" was used... why not state that instead of "adequate sample size"?

Response: We have adjusted the sentence in line 372.

Comment 28: Line 366: is it "prevalence of PPD" or "risk of PPD" or "prevalence of risk of PPD"???? Please be consistent all through the manuscript...

Response: Thank you for pointing this out. We have adjusted the manuscript to make this clearer in numerous sections. 

CONCLUSION

Comment 29: Line 381: Should the initial part of the sentence not be "the prevalence of PPD" or "risk of PPD at an EPDS score of 9 or more"???? Thought it was the "prevalence of PPD being measured with the EPDS" and "not the prevalence of EPDS".

Response: We have adjusted the sentence in line 394-396 to be more precise.

Comment 30: Line 384: In place of "had a substantial impact", why not state clearly that "there was a drop in prevalence rates with higher cut-offs for EPDS"??? That would be much more to the point

Response: We have rephrased the sentence in line 397-399 accordingly.

---

## [Editor Report · Decision Letter 2]

9 May 2022

Mothers at risk of postpartum depression in Sri Lanka: A population-based study using a validated screening tool

PONE-D-21-20712R2

Dear Dr. Solås,

We’re pleased to inform you that your manuscript has been judged scientifically suitable for publication and will be formally accepted for publication once it meets all outstanding technical requirements.

Kind regards,

Yeetey Akpe Kwesi Enuameh, MD, MSc, DrPH

Academic Editor

PLOS ONE

Additional Editor Comments (optional):

The authors have addressed the issues raised. Just one final suggestion, could you replace "crosstabulation" on line 26 of the abstract with "univariate" and change "regression" to plural?

Congratulations and thanks for your patience.